# An Ensemble Approach for the Prediction of Diabetes Mellitus Using a Soft Voting Classifier with an Explainable AI

**DOI:** 10.3390/s22197268

**Published:** 2022-09-25

**Authors:** Hafsa Binte Kibria, Md Nahiduzzaman, Md. Omaer Faruq Goni, Mominul Ahsan, Julfikar Haider

**Affiliations:** 1Department of Electrical & Computer Engineering, Rajshahi University of Engineering & Technology, Rajshahi 6204, Bangladesh; 2Department of Computer Science, University of York, Deramore Lane, Heslington, York YO10 5GH, UK; 3Department of Engineering, Manchester Metropolitan University, Manchester M1 5GD, UK

**Keywords:** diabetes mellitus, artificial intelligence (AI), machine learning (ML), explainable AI, ensemble classifier, soft voting

## Abstract

Diabetes is a chronic disease that continues to be a primary and worldwide health concern since the health of the entire population has been affected by it. Over the years, many academics have attempted to develop a reliable diabetes prediction model using machine learning (ML) algorithms. However, these research investigations have had a minimal impact on clinical practice as the current studies focus mainly on improving the performance of complicated ML models while ignoring their explainability to clinical situations. Therefore, the physicians find it difficult to understand these models and rarely trust them for clinical use. In this study, a carefully constructed, efficient, and interpretable diabetes detection method using an explainable AI has been proposed. The Pima Indian diabetes dataset was used, containing a total of 768 instances where 268 are diabetic, and 500 cases are non-diabetic with several diabetic attributes. Here, six machine learning algorithms (artificial neural network (ANN), random forest (RF), support vector machine (SVM), logistic regression (LR), AdaBoost, XGBoost) have been used along with an ensemble classifier to diagnose the diabetes disease. For each machine learning model, global and local explanations have been produced using the Shapley additive explanations (SHAP), which are represented in different types of graphs to help physicians in understanding the model predictions. The balanced accuracy of the developed weighted ensemble model was 90% with a F1 score of 89% using a five-fold cross-validation (CV). The median values were used for the imputation of the missing values and the synthetic minority oversampling technique (SMOTETomek) was used to balance the classes of the dataset. The proposed approach can improve the clinical understanding of a diabetes diagnosis and help in taking necessary action at the very early stages of the disease.

## 1. Introduction

### 1.1. Diabetes-Facts and Figures

Diabetes related diseases have recently become one of the top ten causes of death in developing countries. The government and individuals are funding research projects to find an easier and faster way to detect the disease at an early stage. There are two types of diabetes: type-1 and type-2. Type 2 diabetes is characterized by high blood sugar, insulin resistance, and a relative lack of insulin. Insulin resistance occurs due to excessive fat in the abdomen and around the organs, which is called visceral fat. The majority of obese individuals have elevated plasma levels of free fatty acids (FFA) which are known to cause peripheral (muscle) insulin resistance [1]. Type-1 diabetes is a condition in which blood sugar levels rise due to a shortage of insulin, causing problems with the blood sugar metabolism. Most food people eat is broken down into sugar (glucose) and released into the bloodstream. When blood sugar levels rise, the pancreas cell releases insulin, which provides energy for everyday tasks [2]. Excessive blood sugar stays in the bloodstream when there is not enough insulin or if the cells stop responding to the insulin. This can cause serious health problems such as heart disease, vision loss, and kidney disease in the long run. The symptoms of diabetes generally are physical weakness, itching, delayed healing, muscle stiffness, polydipsia, and visual burring [3]. Diabetes is a metabolic condition that results in millions of deaths each year throughout the world due to a variety of health complications. By 2030, the number of people with diabetes in developing countries is expected to rise from 84 million to 228 million [4] imposing a significant load on every healthcare system around the world [5].

### 1.2. Problem Statement

Diabetes is the reason for the change in glucose levels in the body. Some preventive measures, such as a balanced diet and a healthy lifestyle, can be considered [6] to reduce the risks of diabetes. Diagnosing diabetes is easier with a regular medical checkup. Laboratory tests are also performed to detect the disease. Patients with the type 1 diabetes require life-saving insulin for as long as they stay alive, though, for the type 2, most of them do not need insulin. This unhealthy situation depletes individuals, families, and national resources if left untreated. An early identification and palliative treatment are essential for prediabetic patients’ health and well-being. An intelligent system based on disease symptoms and laboratory tests will be helpful in the diabetes diagnosis and prevention.

### 1.3. Artificial Intelligence (AI) Research Challenges in a Diabetes Diagnosis

AI has been used for disease diagnosis for several years. It offers excellent outcomes for detecting different types of diseases [7,8], forecasting the pandemic outbreak in any country or region [9], and for many other applications. An intelligent ML-based diagnostic method can correctly detect diabetes at an early stage. For identifying the presence and absence of diabetes with a ML-based system, an appropriate dataset with relevant features for training is essential. There has been much research carried out on the diagnosis of diabetes. Some research has obtained poor accuracy [2,10,11] because of an inappropriate model selection and a lack of data preprocessing. Although, some research has provided a better performance in terms of accuracy, the explanations behind the decision have not been described adequately [12,13]. Therefore, both reliable and explainable AI models are required in the medical area for a better interpretation of the model output and easily comprehensible by the medical and health professionals.

### 1.4. Research Motivation

Pima is a very well-known diabetes dataset and recent work based on this have shown good results. Since the dataset has a high variance, it is easy to achieve an inflated accuracy using a train test split approach. However, it still remains a challenge to achieve a reliably high accuracy using the cross-validation (CV) technique. Furthermore, the class of the dataset is imbalanced, and this requires applying a proper class balancing technique to avoid any overfitting or underfitting problems. Finally, the lack of explainability of the existing ML methods, diabetes detection, and progression prediction still remain a challenging area despite the significant current research efforts.

Although diagnosing diabetes has notably improvement in recent studies, unfortunately, all of the earlier research was concentrated on enhancing the model’s performance while ignoring the interpretability challenges. As a result, despite this research making significant breakthroughs in the disease prediction, they are unlikely to be accepted by the medical community. The necessity of explaining the black box model and making it understandable to everyone motivated the authors to develop efficient and interpretable models for the diagnosis of diabetes.

### 1.5. Aim, Contribution, and Paper Organization

This research intends to provide an interpretability of the ML models and enhance the prediction performance with data pre-processing in order to diagnose diabetes. For the preprocessing, the data was scaled for some algorithms, the missing values were replaced using median and class imbalance, and were handled using the SMOTETomek. Thus, the used models would offer an effective, reliable, and explainable diabetes prediction.

The major contributions of this work are listed in the following bullet points.

Several machine learning algorithms were applied and using the two best classification methods, an ensemble method was developed to diagnose diabetes.The model’s inside explainability was provided to make the model more reliable and to produce a good balance between the accuracy and interpretability, which will be convenient for doctors or clinicians to understand and apply the model.SHAP plots were created to provide physicians with some insights into the main driving factors affecting the disease prediction from various perspectives, including visualization, feature importance, and each attribute’s contribution to making a decision.

The rest of the paper is arranged as follows. The current literature is reviewed in Section 2. The proposed approach along with the description of the datasets and algorithms is presented in Section 3. The model’s performance with a brief explanation behind the decision are presented in Section 4. Finally, Section 5 concludes the key findings of the work and outlines the important directions for future work.

## 2. Literature Review

Researchers have been experimenting with various ML approaches to predict diseases as early as possible. Various ML algorithms, particularly hybrid techniques, have been developed to improve the model outcomes. Several researchers have used the Pima Indian diabetes dataset (PIDD). Some of the related works are discussed here.

Different types of ML algorithms have been used in [6,12,13,14,15,16,17] for the diagnosis of diabetes. Since the PIDD has imbalanced classes, the preprocessing classes need to be balanced. In these studies, they did not balance the class of the PIDD except for [17]. That’s why the models were biased toward the majority class. In [14], for the PIDD, missing values were replaced by the mean values. Then, the incorrectly classified data was removed using the k means clustering algorithm. A decision tree classifier was used for the classification using the reduced dataset. For the same dataset, other researchers [15] used several machine learning algorithms and the support vector machine (SVM) performed better than the others, with an accuracy of 78.20% (using 70% of the data for the training). Here, accuracy is the number of correctly predicted diabetic and non-diabetic patients from the records of all of the patients. This accuracy is comparatively poor comparing others’ accuracy using the train test split ratio. The models were built using three machine learning algorithms with the PIDD dataset [16], where the SVM provided the highest accuracy of 80 with 70% of the data used for the training. The same dataset was also used by Tiwari and Singh [12] and a decent accuracy of 78.9% was obtained by the XGBoost classifier. In another study [13], researchers proposed a ML-based e-diagnosis system and investigated the algorithm’s performance using a variety of fine-tuned features. For the binary classification, the naive Bayes model appeared to perform well with a fine-tuned selection of features, whereas the random forest (RF) model did better with additional features. A vast difference was observed between sensitivity and specificity because of the imbalanced class. Kibria et al. [6] found an accuracy of 83% in diabetes detection using the logistic regression (LR) where the KNN algorithm was employed for the imputation of the missing values. By using the appropriate process to replace the missing values and balance the data distribution, opportunities can be created to improve the prediction performance.

There is room for improvement in the preprocessing steps of these studies. The appropriate approach to handle a lot of the missing values and imbalanced classes resulted in a poor accuracy. In these works, the authors did not use any hybrid method or ensemble learning for further improvement in the performance metrics. A dataset with imbalanced classes could be the cause of the poor outcomes. Balancing the dataset, taking the necessary steps to replace the missing values and select suitable algorithms will help the model to predict more accurately. The drawback of the class imbalance was solved by Ramesh et al. [17] by applying the SVM-RBF kernel for the classification, the SMOTE technique for balancing the dataset, and some feature selection techniques for extracting the characteristic features. Using a ten-fold stratified cross-validation approach, this study attained an accuracy of 83.20%, a sensitivity of 87.20%, and a specificity of 79%.

The ensemble models used in [10,18,19,20], showed a better performance than any single ML algorithm, that is why the usage of ensemble models to diagnose disease has increased. In [21], the researchers built a decision-level fusion model to predict heart disease, which was further improved by applying the weighted score fusion [7]. The ensemble method was used in the diabetes detection and found promising results. The ensemble technique was also applied by Kumari et al. [10], who obtained an accuracy of 79.04% by applying a soft voting classifier for diagnosing diabetes mellitus. The efficiency of the ensemble soft voting classifier was tested using a breast cancer dataset, where an accuracy of 97% was obtained. For preprocessing, they used a min-max normalization, and for the missing values, the median of the attribute was used. Fuzzy logic with the fusion model was utilized in [18]. Two types of models, the SVM and the ANN, were combined for the classification. The results of these models became the fuzzy model’s input membership function, and the fuzzy logic determined whether a patient had diabetes or not. 94.87% accuracy was obtained using 70% of the data for training which proved to be better than the earlier studies. However, the interpretability of the ML models was not shown. Abdollahi and Nouri-Moghaddam [19] introduced the stacked generalization and they used an ensemble approach with a genetic algorithm to diagnose the disease with a promising outcome. An accuracy of 98% was achieved using 70% of the data for training.

The lack of preprocessing and imbalanced classes in the dataset resulted in a poor accuracy in these studies [10,18,19]. These limitations were overcome by Fitriyani et al. [20] who developed an ensemble model to predict the diabetes disease. The outliers were removed using the isolation forest, and the class of the data was balanced using the SMOTETomek. The model built using four datasets provided an accuracy of 96.74, 85.73, 75.78, and 100% for diagnosing diabetes (dataset-1, 17 features) and hypertension (dataset-2,3) though the interpretability of the model was missing. The last dataset was used to find the relation between diabetes and hypertension. They did not use the Pima dataset. Finally, they created a smartphone application for real-world use.

The above-mentioned works achieved a better output in terms of accuracy, but the explanation behind the decision was not discussed. The contribution level of each feature behind the prediction of a decision was not explained, hence making it difficult to understand how a decision was made by the model. Most of the works [6,10,12,16,19,22] used the train test split for diagnosing the diabetes disease, which resulted in a high accuracy. For example in [19], the accuracy was 98% for an imbalanced class of dataset where training, validation, and testing the amount of data was 70%, 15%, and 15% respectively. Furthermore, an accuracy of 83% was obtained using 70% of the data for training with an imbalanced class in [6]. The variance of the Pima dataset is very high and using a train test split ratio will never return a true accuracy, rather it will give a biased accuracy which is possible that the model is only giving this high accuracy for only a particular set of the training data. Moreover, the training accuracy also was not reported, therefore no option is available to verify whether the model was under-fitting or not.

Unfortunately, all of the earlier research concentrated on improving the model’s performance while ignoring the interpretability challenges. As a result, despite these studies making significant breakthroughs in the disease prediction, they are unlikely to be accepted by the medical community. Therefore, a notable gap exists between the academic research findings and their effective application in medical practice due to multiple reasons [23]. Furthermore, despite their great accuracy, physicians frequently do not rely on the most up-to-date techniques and methodologies [24]. Most of these approaches and methods are fundamentally non-transparent, difficult to understand, and unable to answer simple questions such as: Why is this conclusion drawn? Why it is essential from a medical standpoint [25]? Patterns discovered from the datasets using complicated ML algorithms are not always accurate or easily understandable. Therefore, the medical specialists do not accept the black-box models that do not provide a thorough and simple explanation [26]. For these reasons, the clinical ML approaches typically avoid complex models in favor of simpler and more interpretable models with the sacrifice of a higher accuracy [27]. Many researchers have attempted to open the black box of sophisticated models and explain their decisions by understanding how they function or by demonstrating their decisions [28]. This emerging approach is known as XAI, which stands for “accountable, transparent, actionable, or explainable artificial intelligence.” The ability of the ML algorithms to explain (mathematically) or predict their outcomes in terms human comprehension is explainability.

Due to the biased accuracy and lack of explainability [2,12,13] of the existing ML methods, diabetes detection and progression prediction still remain a challenging area, despite the significant current research efforts. This research intends to provide an interpretability of the ML models and enhance the prediction performance using the cross-validation with the data pre-processing to diagnose diabetes. Thus, the used models would offer an effective, reliable, and explainable disease prediction.

## 3. Methodology

### 3.1. Proposed Approach

The whole workflow of the proposed approach is demonstrated in Figure 1. The data was downloaded from Kaggle (https://www.kaggle.com/datasets/uciml/pima-indians-diabetes-database, accessed 10 September 2022) then it was cleaned and pre-processed (missing values imputation, class balance, etc.). Following the preprocessing, the stratified data was five-fold cross-validated. The class of the data was highly imbalanced. The SMOTETomek algorithm was used to balance the classes after CV for each fold. It should be noted that the SMOTETomek was only applied to the training dataset. The class of the test set was not balanced. The class imbalance with the CV should be handled after the train test split, since if the class balancing was produced before the cross-validation, it will affect the test set. Therefore, the appropriate way to use the CV with a balanced class of dataset is to balance the class after the CV. Five isolated folds were generated from the training set using the CV and then the SMOTETomek was applied to each isolated fold. Consequently, each training fold produced a balanced class of the training fold, but the test fold remained the same. This CV method has also been discussed in [22]. Following the balancing of the class of the data, six ML algorithms were applied: artificial neural network (ANN), random forest (RF), AdaBoost classifier (ADA), XGBoost classifier (XGB), support vector machine (SVM), and logistic regression (LR) were used for training. Based on the performance of the algorithms, the two best-performing algorithms were selected from the six ML algorithms and a weighted ensemble model was developed for the diagnosis of diabetes. Different weights have been selected for every fold with XGB, and the AdaBoost order, respectively. A soft voting classifier was used to develop the ensemble model. Following the training, all trained models were used to predict the test data. Here, the LIME package and the SHAP tool were used for the model explanation. The trained ML algorithms and the ensemble model are explainable supervised algorithms. Both the global and local explanations were described using those algorithms.

### 3.2. Dataset Description

The Pima Indian dataset (https://www.kaggle.com/datasets/uciml/pima-indians-diabetes-database, Accessed 10 September 2022), used in this work consisted of a total of 768 instances with nine attributes of diabetes detection to formulate a binary classification problem. This dataset contains information about female patients only. Table 1 displays the description of the attributes, and Table 2 presents the statistical values of the dataset. This dataset was selected because this is a very common publicly available dataset to predict diabetes, and most researchers employed this dataset to develop models. Therefore, it would be convenient to compare the proposed model with others and identify the space for improvement.

Every feature’s seaborn plot is displayed in Figure 2. The relationship between one feature with respect to the other eight features, including itself, has been plotted. This plot is helpful in identifying the relationships of the features. If the points are scattered, there is no absolute relationship, while if the points are approximately placed in a straight line, they show a linear relationship between them. While referring to the seaborn plot, insulin vs. glucose and skin thickness vs. BMI are the most positively correlated features.

The Pearson correlation heatmap between all of the features is displayed in Figure 3. It is calculated based on the value of two features and measures the linear relationship between them. The correlation between the two features was measured using the Pearson correlation. The correlation coefficient values ranged from −1 to +1. A value closer to 0 implies a weaker correlation. 0 means no correlation. A value closer to 1 or −1 indicates a stronger positive or negative correlation, respectively. The strongest positive correlation was found between the BMI and skin thickness. While age vs. pregnancies and glucose, vs. insulin also showed a positive correlation. There is no noticeably strong negative correlation between the features. The Pearson correlation coefficient was calculated after the missing value imputation.

### 3.3. Preprocessing

The first work for any data mining technique is data preprocessing. It plays a vital role in model performance [31]. The dataset contained a lot of missing values and the class of the data was imbalanced.

#### 3.3.1. Missing Value Imputation

The missing values in the dataset are visible in Figure 4 where insulin had the most missing data compared to any of the other features. Other features, including skin thickness and pregnancy, were also lacking. Here, some features had zero values which do not make any sense. These values were treated as missing values in the dataset. The features where zero was treated as missing values were glucose, blood pressure, skin thickness, insulin, and the BMI. These zero values were replaced by “NaN” in the dataset. To replace the missing values of the features, the median was taken corresponding to the target value. Since this was a binary classification, each feature had two medians for two classes. For instance, the median for glucose was 107 and 140 for the two classes (Normal and diabetic patients), respectively. The missing values of glucose were replaced by 140 for the diabetic patients and for the normal patients, it was replaced by 107.

#### 3.3.2. Data Partitioning

The entire dataset was stratified, and then a five-fold cross-validation was applied to the dataset.

#### 3.3.3. Handling Imbalanced Classes of a Dataset

There are several methods to make a dataset balanced. In this work, the SMOTETomek, a combination of the SMOTE and the Tomek algorithms, was applied. The SMOTE is an acronym for synthetic minority oversampling technique. Tomek is an undersampling technique. First, the SMOTE was applied to create new synthetic minority samples to obtain a balanced distribution of the classes. Furthermore, the Tomek link was used to remove the samples close to the boundary of the two classes in order to increase the separation between the two classes [32]. It was applied only to the training dataset and the test set remained the same. Table 3 represents the data distribution of each class before and after using the SMOTETomek on the training dataset.

#### 3.3.4. Feature Scaling

A min-max normalization was used on the dataset except for the tree-based algorithms such as the random forest, the AdaBoost, and XGBoost classifiers. The SVM, logistic regression, and the ANN algorithms need normalization. The min-max scaler is defined by Equation (1).
(1)$$h^{\prime }=\frac{h-\min\left(h\right)}{\max\left(h\right)-\min\left(h\right)}$$
where *h* is the original value, and *h*′ is the normalized value.

#### 3.3.5. Weighted Score Approach for the Ensemble Method

For the ensemble classification, a weighted model was developed. Two weights were assigned to the two algorithms used in the ensemble approach. For every fold, a loop was used to check a combination of weights producing the highest accuracy, which was selected for every fold. Since it was a five-fold cross-validation, the weights were updated in each fold.

### 3.4. Models and Algorithms

#### 3.4.1. Ensemble Learning

The individual models were combined in the ensemble approach to improve the model’s stability and predictive power [33]. This approach permits a higher predictive performance compared to a single model. The ensemble finds ways to combine multiple machine learning models into one predictive model. Bagging is used to reduce variance, boosting reduces bias, and stacking improves performance. Specific models do well in modeling one aspect of the data while others do well with another aspect. Instead of learning a single complex model, the ensemble model learns several simple models and combines their outputs to produce the final decision. The combined strength of the models offsets the individual model variances and biases. The ensemble learning will provide a composite prediction where the final accuracy is better than the accuracy of the individual classifiers. The weighted soft voting approach has been used in the proposed method and the equation is given below:
(2)$$\hat{y}=arg\;\underset{i}{\max}\sum_{j=1}^{m}w_{j}p_{ij}$$

Here, *p* is the predicted probability for each classifier, and *w_j_* is the weight given to the *j*th classifier.

#### 3.4.2. AdaBoost

The ensemble method is divided into two groups, the sequential or bagging technique, and the parallel or boosting technique. In sequential ensemble methods, base learners are generated consecutively. AdaBoost is a sequential ensemble method. The basic motivation of the sequential methods is to use the dependence between the base learners by weighing the previously mislabeled examples with a higher weight. Therefore, the overall performance of a model can be boosted. Bagging combines the results of multiple models to obtain a generalized result from a single model. Bagging reduces the variants of an estimate by taking the mean of various estimates [34].

#### 3.4.3. Random Forest

The RF is a boosting technique. The parallel ensemble methods are applied wherever the base learners are generated, in parallel. Each base learner model is provided with a sample of data; these base learners give the output individually. At last, based on the base learners’ predictions, the final prediction is made based on the voting classifier. The RF builds multiple decision trees and merges them to obtain a more accurate and stable prediction. In the RF, the base learner models are decision trees. Since the errors are often reduced dramatically by averaging, the basic motivation of the parallel methods is to use independence among the base learners.

#### 3.4.4. XGBoost

XGBoost repeatedly builds new models and combines them into an ensemble model. First, from one developed model, the error of the residuals for each observation is calculated. Based on the prior errors, a new model is created to anticipate those residuals. Then predictions from this model are added to the ensemble models. Compared to gradient boosting algorithms, XGBoost is preferable because it achieves a fair balance of bias and variance.

#### 3.4.5. Logistic Regression

The LR is a linear regression transformation algorithm that allows for describing binary variables in a probabilistic manner. It is a classification algorithm that is used to find a relation between attributed and a particular outcome’s probability. The logit function is utilized in this classification method, hence the word “Logistic.” It is highly useful in medical diagnoses, given some particular symptoms and characteristics. Like other regression analyses, the LR is a type of predictive analysis. It calculates the outcome’s predicted probability. It is a particular case of linear regression with a categorical target variable. In a logistic regression, the effect of the outlier is removed by adding the logit function [34].

#### 3.4.6. Support Vector Machine

A support vector machine (SVM) is a linear model for classification and regression problems. Both linear and nonlinear problems can be solved using it. It works in a similar way as a linear regression. In the SVM, the algorithm classifies new data by generating a hyperplane with a maximum marginal distance.

#### 3.4.7. Artificial Neural Network

The ANN has three main layers- the input, hidden, and output layers. The input is given in the input layer, and the output is received from the output layer. The middle layers are for adjusting the weight and reducing the error between the true value and the target value. This process is known as backpropagation [35]. In the proposed ANN, the number of nodes in the input layer was eight, and there are two middle layers containing nodes ten and eight, respectively. Since it is a binary classification, there is only one node at the output layer. Twenty epochs were used for each cross-validation. For the first two layers, the ReLu was used as an activation function, and the sigmoid was used for the last layer.

The hyperparameters of any algorithm must be tuned to obtain the best result for any dataset. The hyperparameters of the algorithms were also tuned to achieve the desired outcome. To select the best performing weights for the ensemble model, the model with some weights were evaluated and then the weights selected were those that produced the best accuracy. Furthermore, not all of the algorithms for the ensemble model were considered. The best two performers were used for the ensemble method among the six algorithms. All of the tuned parameters have been shown in Table 4.

#### 3.4.8. Reproducible Models

During the development of any CV method, it is useful to be able to obtain the reproducible results from run to run and to determine if a change in the performance has happened due to an actual model or data modification, or is merely a result of a new random seed.

Since some algorithms are irreproducible, such as the RF and the ANN, the only way to ensure that the results of these models are reproducible is to set a quantity known as the random seed, which controls how random numbers are generated. Thus, the models become reproducible. Therefore, the random seed was set for every algorithm used in this study. For the ANN, the random seed was set to a specific value. For other ML models, in sklearn, a parameter named “random state” was used to control the randomness. By setting the parameters, the ML algorithms took the defined combination of the seeds in every fold. Therefore, the proposed ensemble approach produced reproducible results.

#### 3.4.9. Shapley Additive Explanations (SHAP)

It can be challenging to justify the model’s reliability when creating complex models. While the global performance matrix such as accuracy is helpful, they cannot be used to describe why a model correctly predicted a particular outcome. The LIME package and the SHAP tool were mainly used to explain and visualize the model. A python package: LIME is a method that is fitted to a local model around the area in question to explain the result of black-box models. It is a game-theoretic method [36] for explaining how machine learning algorithms reached their decision [37]. SHAP is a visualization tool that explains any machine learning algorithm by visualizing its outcome. SHAP computes the contribution of each feature in a dataset to the prediction. Thus, the explanation of any model can be described by SHAP. It combines some other tools, such as LIME and many more [38]. The SHAP values have become very popular in explainable AI and are also used in feature selection [39], and model explanation [40]. In this study, LIME and SHAP were used to explain the proposed approach.

However, extensive computational time is a challenge for SHAP and this time depends on the SHAP explainer. There are different types of SHAP explainers available. Kernel and tree explainers were used in this work. Among them, the tree explainer works with all tree-based algorithms and is much faster compared to the kernel explainer. To calculate the SHAP values of the ANN, the kernel explainer was used and it was a very time-consuming method. Though the kernel explainer is a universal explainer (it works with any algorithms), because of its high computational time, the tree explainer was used in the proposed work, for the tree-based algorithms.

A common way to understand the influence of each feature of the ML models is to examine the coefficients learned for each feature. From those coefficients, it can be understood how the model output changes with a change in the input feature. However, it is not a reliable method to measure the overall importance of a feature as the coefficients depend on the scales/units of each feature. If the scale of the feature changes, the coefficient also changes. Therefore, the magnitude of the coefficient is not a good choice to understand the importance of the feature of a model [41]. In this case, SHAP is a perfect choice to see the individual as well as the overall influence of the features. Furthermore, using this coefficient method, local explanations cannot be observed but this can be easily achieved using SHAP.

## 4. Performance Analysis and Experimental Results

Six machine learning algorithms have been applied to the Pima Indian dataset. Some prepossessing was carried out on the dataset before applying the algorithm. Following the application of the six ML algorithms to the preprocessed dataset, an explainable weighted ensemble method was developed, based on a voting classifier. The proposed approach used soft voting, and the weights were selected for the ensemble method.

### 4.1. Performance Parameter

For the classification of the diabetes disease, five quality parameters have been calculated. The performance parameters are given below:(3)$$\mathrm{Accuracy}\;=\frac{T_{P}+T_{n}}{T_{P}+T_{n}+F_{p}+F_{n}}$$
(4)$$\mathrm{Precision}\;=\frac{T_{P}}{T_{P}+F_{p}}$$
(5)$$\mathrm{Recall}\;=\frac{T_{P}}{T_{P}+F_{n}}$$
(6)$$F\text{1-score }=\frac{2*Precision*Recall}{Precision+Recall}$$
where *T_p_* is true positive, *T_n_* is true negative, *F_p_* is false positive, and *F_n_* is false negative. The accuracy is defined as the fraction of all of the correctly predicted diabetic and non-diabetic patients out of the records of all of the patients. The precision is the fraction of the correctly predicted diabetic patients out of all of the correctly and incorrectly predicted diabetic patients. The recall calculates the fraction of correctly predicted diabetic patients out of the records of all of the true diabetic patients only. The F1 score measures the weighted score of the precision and the recall.

### 4.2. Performance Results

This section shows the performance of all of the algorithms considered here. The five-fold stratified cross-validation was used for the algorithms, and at last, an ensemble model was developed using the best two algorithms. The soft voting technique was used for the ensemble method.

The ANN produced an accuracy of 79%, as displayed in Table 5. In addition to handling big data sets, the ANN can implicitly discover the complicated nonlinear correlations between the dependent and independent variables and the possible interactions between the predictor variables. The dataset had a lot of nonlinear relationships shown in Figure 2. However, the ANN did not perform very well for this dataset. In fact, the tree-based algorithms did better than the ANN. The ANN is a complex algorithm and generally works well on large datasets with lots of features. Since the used dataset was not large enough and only had eight input features, the tree-based algorithms outperformed on the dataset.

The SVM provided an accuracy of 79% and an F1 score of 80%, as shown in Table 5. The SVM is also one of the best classifiers for the binary classification problems with the balanced class of datasets, free or with little noise. Since the dataset used here was not outlier-free, that is why the SVM did not perform well with this dataset. The LR provided an accuracy of 78%, with an F1 score of 78%. Other than the ensemble, the XGB showed the most promising results among the six ML algorithms. Both the XGB and the RF faced the overfitting problem. The XGB is a greedy algorithm that can over fit a training data quickly. Regarding the RF, if the hyperparameters were tuned to the maximum depth, then the accuracy also decreased. Therefore, the sklearn default parameters were used for the RF. The performance metrics of all of the algorithms for every fold can be found in Appendix A.

From Table 5, it could be observed that the tree-based algorithms such as the RF and the XGB produced the same accuracy of 88%. In the case of diabetes, it cannot be ruled out that if a particular situation is present, then that patient must have diabetes, many other relevant issues could cause diabetes. Of course, there are some patterns, such as in many cases, if there are high glucose levels or insulin, there is a good probability that the patient may have diabetes. However, these conditions are not always relevant to the output. And the tree-based algorithms performed better when not every condition was relevant to the action. For this reason, the RF and the XGB performed well. AdaBoost is also a tree based algorithm and it performed better than the ANN, SVM and the LR, by providing an accuracy of 83% for the test data.

The Is also a tree based algorithm The voting classifier was combined with the RF and XGBoost. Since the performance of these two algorithms was better compared to the others, they were selected for the voting classifier. The weighted voting technique and performance are also shown. Here, the voting classifier provided an accuracy and F1 score of 90 and 89%, respectively, and these outperformed all of the algorithms.

Figure 5 presents the ROC curves of each fold for all of the algorithms used. The AUC scores of the XGB and the voting classifier was the highest (95%), while the AUC score of the ANN and the LR were comparatively lower (87% and 86%, respectively) than the others.

### 4.3. Comparison with Previous Research

Other authors used various methods to classify the same PIMA dataset used in this study and achieved a decent accuracy. Table 6 presents the latest work with the PIMA dataset only. Most of them carried out basic preprocessing such as scaling, and label encoding. An important drawback of these studies was that they did not balance the class of the dataset. The class of the Pima dataset was highly imbalanced, which might result in a biased accuracy. To check the performance of the algorithms, other performance metrics are necessary. To solve the problem, the SMOTETomek was used in the proposed approach to avoid any overfitting. Six ML algorithms were applied on the dataset and the soft weighted ensemble approach outperformed all of the algorithms. The proposed model provided an accuracy of 90% where both recall and F1-score were 89%.

A comparatively better accuracy was achieved by Chang et al. [13] than the other studies with a 70:30 split ratio where the features were selected using PCA, k means clustering, and the importance ranking to remove the noise of the dataset. The performance was measured using all of the features of the dataset, then a comparison was carried out using the selected three and five features. The decision tree, the random forest, and the naive Bayes algorithms were used for the classification. Using only three and five selected features did not improve the output performance much. Without the feature selection, the RF provided an accuracy of 79.57% with a very poor specificity of 75%. The imbalance of the samples of class 0 and class 1 is most likely responsible for the significant difference between the sensitivity and the specificity. Without using any feature selection method, the proposed model in this paper provided an accuracy of 90%.

Now, for the output performance (where all of the features were used), from the RF, the highest precision and the F1 score were 89.40 and 85.13%. Two reasons could be factored for this higher outcome. First, the class of the dataset was not balanced, therefore, the precision was not a proper performance measure as it was prone to give a higher value, drastically decreasing the specificity. Secondly, the train test split was used for the classification. In a train test split, the performance score depends on how the data is split and the outcome varies significantly for every split. Multiple train test splits could be carried out to reduce the biased result, which was not considered. Furthermore, the cross-validation assures an unbiased result. Since the PIMA dataset’s variance is high, the train test split produced a significantly biased result. That is why the F1 score was much higher than the other approaches. The highest F1 score was 85.17% from the RF.

A soft voting ensemble classifier was used by Kumari et al. [10] and achieved an accuracy of 79%. Since the class of the dataset was not balanced, the models produced a poor performance in terms of the precision, the recall, and the F1 score. The missing values were also replaced by using the median of the specific attribute. This could also decrease the performance. When using a median, it must be different for the target classes. Furthermore, for the PIMA dataset, not all of the zeros should not be treated as missing values. These issues needs to be taken care for in order to achieve a better outcome. Other than the ensemble method, other machine learning algorithms did not perform well.

In [2], the ANN, the RF, and the K-Means clustering were used to examine the diabetes dataset. The highest accuracy was 75.70% from the ANN. A dataset of the imbalanced classes was used for the classification. The same drawback was also found in the work proposed by Tiwary and Singh [12], where the result of the dataset (imbalanced class) affected the prediction performance with a good accuracy (78%), but a very low sensitivity (59%). The recursive feature elimination was selected, which might cause overfitting and result in poor outcomes.

In [11], there is a scope for further improvement in the data preprocessing. A 10-fold cross-validation was used, which provided an unbiased result, and the performance was very poor because of the class imbalance in the dataset. The accuracy can be improved by reducing balancing the dataset and applying preprocessing techniques.

### 4.4. Model’s Explainability

No other previous research on Pima diabetes explains the decision of the model predictions. In this work, the importance of every feature has been determined, and the impact of the features behind a particular decision has been explained. Here, both global and local representations have been shown. The local explanation demonstrates which features contribute the most to a particular test set. The LIME explanation and the SHAP force plot of a test set are the local explanations. In the global explanation, the contribution of the features for a group of data (such as all test data) has been shown. They are the permutation importance of the features, the summary plot of the violin distribution, the SHAP dependence pot and the SHAP force plot with all of the test data.

A test sample with contradictory symptoms was selected for the local explanation. Since this sample was confusing to predict, some algorithms failed to provide the correct result because of the contradictory symptoms. Here, most models said that the patient had diabetes, and the proposed ensemble model’s decision was also predicted the same, correctly. That is why an ensemble model is better performing than the others. The voting classifier favors the right decision in such cases, since the majority generally tells the right. Using the explainable AI, made it convenient for physicians to decide whether the models were performing accurately or not.

#### 4.4.1. Explainability of the Outcome using LIME (Local)

Figure 6 presents the LIME plots for every feature’s positive and negative impacts on making that decision, and physicians can easily understand if the model is not making the right decision. Even if the impact of some features was confusing for the physicians to reach a decision, they could rely on the ensemble method. The positive effect of not having diabetes has been shown in blue, and the features in orange mean that the patient might have diabetes. Here in Figure 6, a single sample was taken to show every feature’s influence. This particular sample was used for all of the algorithms. Since in the ANN, SVM, and the LR scaled data needed to be used, the representations had been shown using the scaled data. If a natural unit was shown, then the LIME explainer needed to be trained on the non-scaled data which would return a different trained model than the used one. Therefore, for better understanding the graphs in Figure 6, the values with natural units of the corresponding features have been shown in Table 7. To understand the values, clinicians need to check the actual feature values of patients (for the algorithms where data scaling is a must). However, the proposed approach showed a natural unit of data in Figure 6g, which can be easily understood by clinicians without confusing with the normalized data.

For the RF algorithm, the glucose value is between 101 to 119, which leads to the decision that the patient may not have diabetes as the glucose level is moderate [42]. The most influential feature in the RF is insulin, which is 142, indicating that the patient has diabetes. Their age is below 40, which leads to the positive decision (orange color), which means that according to the value of the age, there is a higher probability that the patient has diabetes. The RF also predicts that the patient has diabetes.

For different test data, SHAP uses different intervals. Since the BMI was 32.50, it took the interval values close to the given value (32.50). If the BMI were 33, then it would take approximately 30 to 35 for the interval. There is no standard to select the intervals. SHAP observed how the value of a particular feature influenced the decision, and according to the observation, final decision is made. It does not follow any particular standard for any feature to decide whether the patient has diabetes or not.

According to the RF, insulin, age, the BMI, and the diabetes prediction function are the most influential features, and glucose, blood pressure, and pregnancies are the most insignificant features. The insignificant parts indicate that the model did not obtain enough information to firmly identify whether the patient has diabetes or not. When the influence of the features is not that significant, the bars representing them are shorter. Only by seeing this figure, physicians can guess if the model is performing correctly since the impact of the prediction will match the facts of medical science. It is a very informative graph to represent the decision of a single test sample.

The output probability of the decision has also been shown in Figure 6. The ensemble model’s feature contribution and the two individual algorithms (RF, XGB) used for the voting classifier, have also been represented here. A particular test set was used for all of the algorithms to demonstrate the importance of the features. Since the data for the algorithms SVM, ANN, and LR are normalized, the figure shows the normalized data. Therefore, to understand the actual value of the features of that test set, the value of the RF or the XGB can be observed where the data normalization was not necessary. Among the ANN, the RF, and AdaBoost, the ANN predicted that the patient did not have diabetes, whereas the other two algorithms suggested that the patient had diabetes. None of the algorithms gave much confidence for that particular sample. Here, the value of insulin, age, and glucose played a significant role in that prediction for the RF and the Ada. Though for the RF and the Ada, the value of glucose, blood pressure, and pregnancies indicate that the patient might have diabetes, the influence of this decision was very little. Both of them predicted that the particular patient had diabetes, based on the impact of the same features, though none of them gave a strong probability.

Moreover, the ANN predicted that there was a 54% probability that the patient did not have diabetes, and the significant features behind the decision were glucose and blood pressure. Here, the value of glucose, blood pressure, and the BMI implied the conclusion that the patient did not have diabetes, but the impact was little. The glucose was 109, which was not very high. Based on the value, the model predicted that there was a good possibility that the patient did not have diabetes. However, in this case, the patient did have diabetes, which the ANN model failed to identify.

In medical cases, it is hard to conclude any decision to see specific symptoms because that can happen due to other effects. Therefore, it is not unusual to make a wrong decision. Why the model is predicting an incorrect decision, or which value of any feature is different, can be visible using explainable AI.

When these models were combined into an ensemble, it gave a 52% probability that the test data belonged to a diabetic patient. Though the probability score decreased compared to the RF and AdaBoost, still the ensemble model gave the correct prediction. The prediction probability is the probability of the prediction for a particular test data produced from the trained algorithm. To obtain the probability, predict_proba in sklearn was used, which returned the probability score for any algorithm.

#### 4.4.2. SHAP Force Plot of a Particular Test Set (Local)

The base value is the average model output for all of the test data if any feature is not known for the current prediction. For example, if 60% of the data from a test set contains the data of diabetic patients, then the base value will be more than 0.50, implying that there is higher possibility for any random test data belonging to a diabetic person. Therefore, the base value is the mean prediction of the test data. The base value and the predicted value are given in Figure 7, which shows that every algorithm’s predicted value and base value differs. For a particular algorithm, the base value will be the same for all of the test data. The horizontal axis indicates the probability of diabetes. For a given test sample, ANN predicted a 54% probability that the patient had no diabetes (Figure 7a).

The influence on the current prediction can be understood by the force plot. The red-colored features positively influence the prediction (tends to increase the predicted value), whereas the blue-colored features have a negative influence (tends to decrease the predicted value). The red-colored features shift the prediction towards the right side (close to 1) from the base value, and the blue-colored features try to shift the prediction towards the left side from the base value (close to 0). For AdaBoost, the base value was 0.47 where, the influence of the blue-colored features (age, BMI, insulin, skin thickness, and pregnancies) is stronger; therefore, the final predicted value shifts towards the left side from the base, concluding the decision that there is a 47% probability that the patient does not have diabetes. Since the probability score is lower than 50%, hence for this test set, AdaBoost predicted that the patient had diabetes. The more the predicted value shifts towards the left side from the base value, the greater the possibility of having diabetes. In Figure 7a, the ANN provided a false negative for this particular test sample, and the glucose and blood pressure features were responsible for that.

To find the numerical influence of each attribute, waterfall plots were used. One random sample was selected to show the force plot and the corresponding water plot to represent the numerical influence (Figure 8). The red color influences the prediction to be positive (approaching towards 1) and blue influences the prediction to be negative (approaching towards 0: the absence of diabetes). In Figure 8b, the most influential feature was insulin with a value of +0.16. It was also found that the numerical influence of age and the BMI was the same but in the opposite direction.

#### 4.4.3. SHAP Force Plot of the Test Set (SHAP Supervised Clustering)

Figure 9 illustrates the supervised clustering of all cases according to their similarities, output values, and features. The hierarchical clustering was used for measuring the similarities. Only the clustering for the ensemble model is shown here for all of the folds and the last fold. They represent the output probability versus the test sample graphs for the 768 test samples (considering every fold) and the 153 test samples (for the last fold). Here, the *x*-axis denotes the test set number, and the *y*-axis is the output probability. The force plot of each test sample was vertically clustered to generate the force plot displayed in the figure for the global explanation. This force plot is clustered based on the similarity of the features.

The output probability with respect to the features (glucose and insulin) and the relative contribution of the individual features considering all folds and the last fold have been presented in Figure 10. The *x*-axis denotes the feature, and the *y*-axis is the output probability. This cluster is based on the output value. The value of glucose ranged from 60 to 180. It is noticeable that with an increase of the glucose value, the quantity of the blue color increased (blue = presence of diabetes, red = non diabetes), both for all folds and the last fold. Not only glucose, but the risk of diabetes also increased with an increase in the insulin level.

#### 4.4.4. Permutation Importance of the Features (Global)

Based on the permutation importance, the importance of a feature can be understood. If a single column of the data is randomly shuffled while the target and all of the other columns remain unchanged, in that case, the change in the accuracy of the model will provide the permutation importance of the shuffled column. A great change in accuracy after the shuffling indicates that the feature is important. If shuffling one column does not have a significant change in the model’s accuracy, then the permutation score of the feature is less. Each model’s permutation importance of the features is represented in Figure 11. According to the algorithms used, glucose was found to be the most influential feature. Other most influential features were pregnancy, age, and BMI, whereas blood pressure and skin thickness were the least influential. This permutation score was calculated using the training sample of the algorithms. Since the five-fold cross-validation was used, the feature importance of every fold was combined to display the final permutation importance.

The summary plot for every algorithm is shown in Figure 12. The impact on a specific class of a specific feature for a given instance is represented by each dot on the plot. The color of the dot represents the magnitude of the contribution to the model impact. The color red denotes a high value, whereas the color blue denotes a low value. Almost all of the glucose red dots were on the left side, indicating that patients with a high glucose value tend to have diabetes. The greater the distance of the dot from the zero position, the greater the impact. The distance of the glucose red dot was the longest on the left side compared to the other features, indicating that it made the greatest impact on the presence of diabetes. The same logic could be applied to the remaining features. It is also medically confirmed that people with a higher glucose level have a higher risk of developing diabetes. Therefore, from this summary plot, it will be convenient for physicians to see whether the model is working correctly. It is also worth noting that age, pregnancy, insulin, skin thickness, and BMI all impacted on the presence of diabetes. If their values are high (red color), the patients might have diabetes.

Another point to notice in Figure 12, is that some red dots are also present on the right side. The XGB predicts some cases where the patient has no diabetes though the blood sugar, glucose, and skin thickness are high. High blood sugar can result from various causes, not just diabetes. Having high blood sugar might increase the risk of developing diabetes. Similarly, people with a high BMI and high blood pressure may or may not have diabetes. Although these are causes of diabetes, it cannot be completely ruled out, if a person exhibits these particular symptoms but is not diabetic. Every symptom is related to other symptoms in order to reach a final decision.

#### 4.4.5. SHAP Summary Plot of the Violin Distribution

The violin plot in Figure 13 provides each feature’s impact and density. It is a global representation of the test set. The color represents the feature value. The red regions mean the feature’s value is high, whereas the blue regions mean a low value. Considering the first feature concerning glucose levels, in Figure 13a, the density becomes narrower towards the left side of the plot, indicating that the patients with higher glucose levels might have diabetes. The number of patients who did not have diabetes, represented by the blue area on the right side, was much denser than left side. Therefore, the high value of glucose on the far edge of the left side of the *y*-axis means they were more prone to having diabetes. Furthermore, for the feature concerning age, the number of patients were high with moderate and low age values, represented by the violet and blue colors, respectively. If the position is on the left side of the *y*-axis, then the patients tend to have diabetes and vice versa. Here, the longer the distance between the vertical line and the position of the feature value, the higher the confidence is in making a positive prediction. As the position of the medium valued age was closer to the *y*-axis, a confident decision could not be made. Therefore, the output probability would not be high, based on the age values that are closer to the *y*-axis. For the feature concerning blood pressure, since the position of the violet color is on both sides, the patients might or might not have diabetes. This feature was the least influential feature in the decision making for the five-fold voting classifier. It should be mentioned that this violin plot was drawn using all of the folds’ test values, as the five-fold cross-validation was used for all of the algorithms.

#### 4.4.6. SHAP Dependence Plot (Global)

Figure 14 presents the SHAP dependence plots based on the ensemble model’s SHAP values with the three most important features (glucose, BMI, and age), according to the permutation importance for all folds. Glucose mostly interacted with the BMI, considering all folds. Furthermore, the BMI interacted with insulin and age interacted with insulin the most.

In Figure 14, red denotes a higher value, whereas blue denotes a lower value. When the value of the BMI increases, insulin also increases, and the probability decreases (*y*-axis). This is the likelihood of a patient not having diabetes. So, for a patient with a high glucose level and a high BMI, that patient has a significantly lower chance of being non-diabetic, thereby leading to the conclusion that the patient has a high risk of having the disease. From this plot, physicians could quickly get the idea of how a patient’s symptoms vary from one to another and how far the patient is from developing diabetes.

## 5. Conclusions

Based on several ML methods, this study provided an accurate and highly explainable ensemble model with the usage of the cross validation approach and by combining two ML algorithms (RF, XGB), using a weighted soft voting classifier to successfully predict the risk of diabetes. It was demonstrated that the predictions based on the weighted ensemble are significantly better than the individual algorithms. The system achieved the highest accuracies by selecting appropriate weights. An accuracy of 90% and a F1 score of 89% was achieved by the ensemble model and was highly competitive with the other models proposed in the literature. The missing value imputation by median values, and the data balancing by the SMOTETomek also contributed to the improved performance. Moreover, the proposed ensemble model produced a favorable accuracy-interpretability trade-off because it achieved accurate results and a high level of interpretability using the permutation importance and SHAP plots. The ensemble model provided logical, medically trustworthy, and practical judgment that can boost a physician’s confidence in real-life implementation.

Even with encouraging findings from an academic perspective, the model is still much farther away from being utilized in a real-world medical scenario, which is intended to be carried out in future. Further studies are needed to assess the performance characteristics of the proposed approach in other relevant datasets. Further improvement in the model’s performance and explainability will be attempted using different ML algorithms to develop different types of ensemble models.

## Figures and Tables

**Figure 1 sensors-22-07268-f001:**
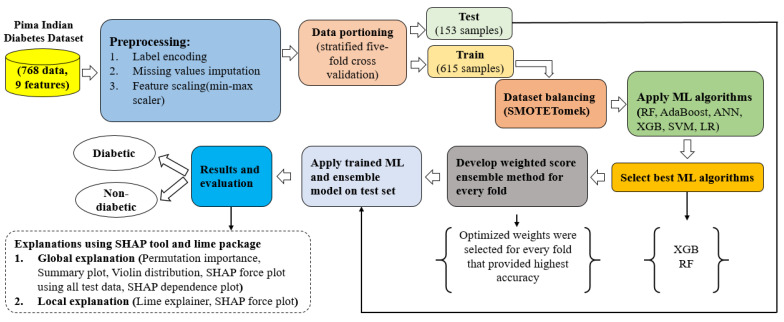
Overall workflow of the proposed diabetes detection model.

**Figure 2 sensors-22-07268-f002:**
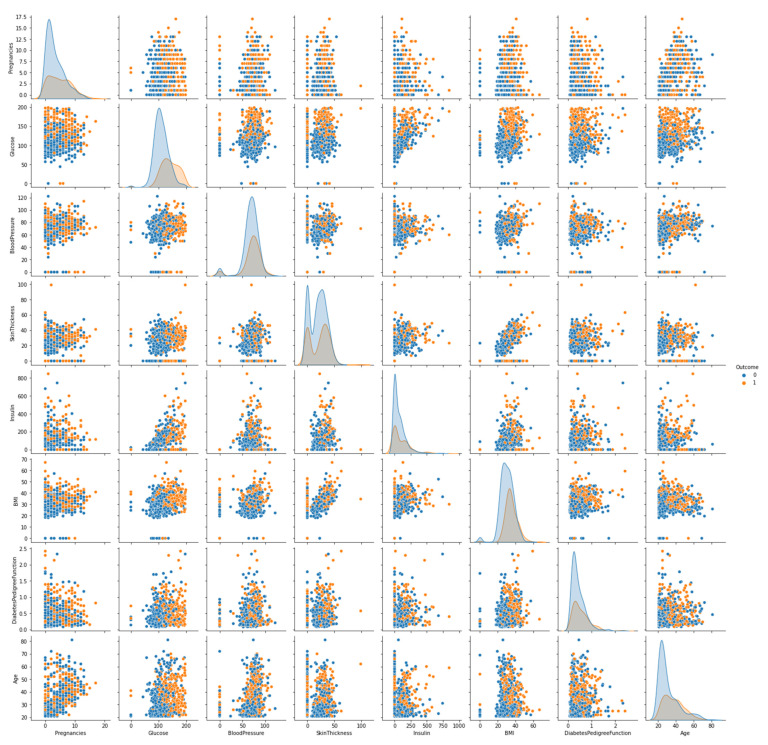
Visualization of all of the attributes present in the Pima Indian Diabetes dataset (outcome 0: non-diabetic and outcome 1: diabetic).

**Figure 3 sensors-22-07268-f003:**
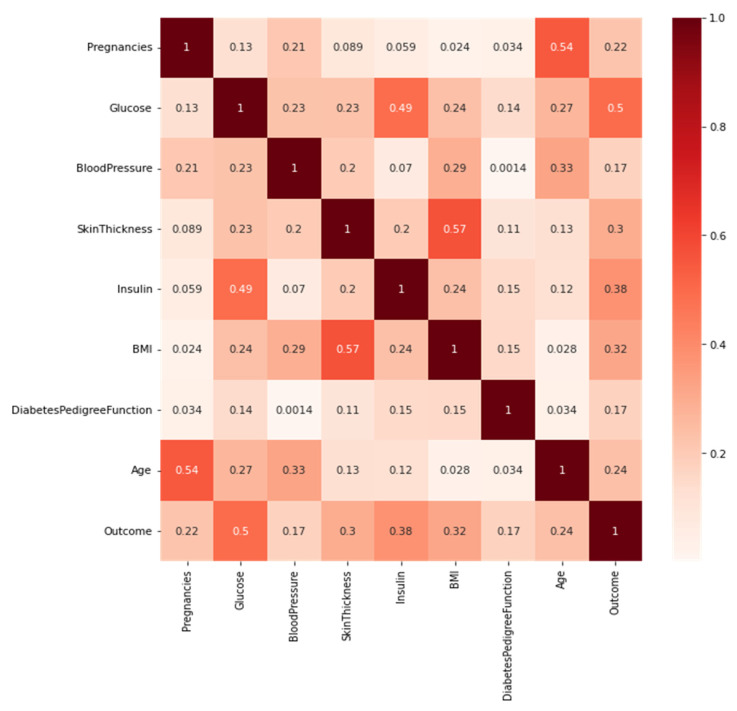
Heatmap of Pearson’s correlation coefficients for all diabetes features.

**Figure 4 sensors-22-07268-f004:**
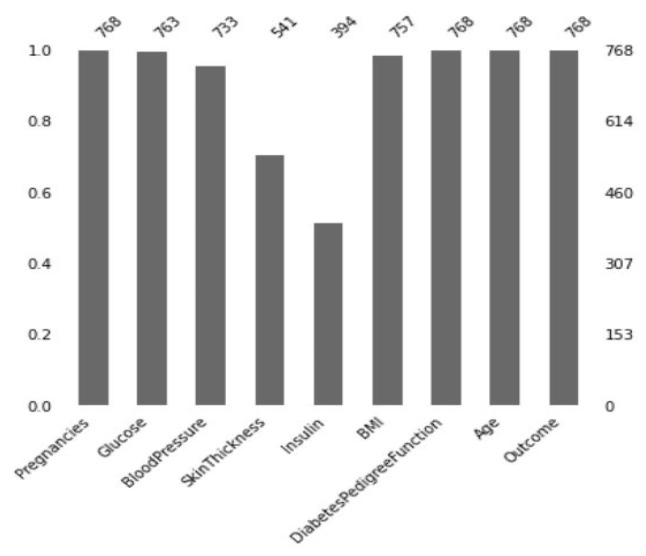
Representation of the missing values before the imputation.

**Figure 5 sensors-22-07268-f005:**
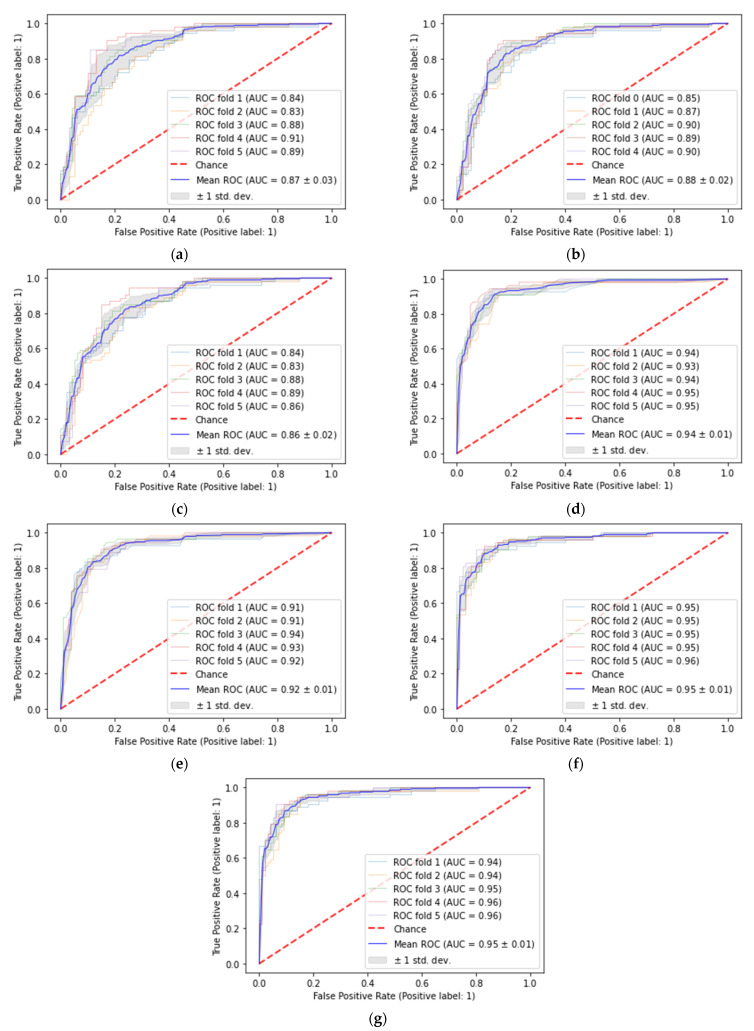
ROC curves of (**a**) ANN (**b**) SVM (**c**) LR (**d**) RF (**e**) ADA (**f**) XGB and (**g**) Voting classifier.

**Figure 6 sensors-22-07268-f006:**
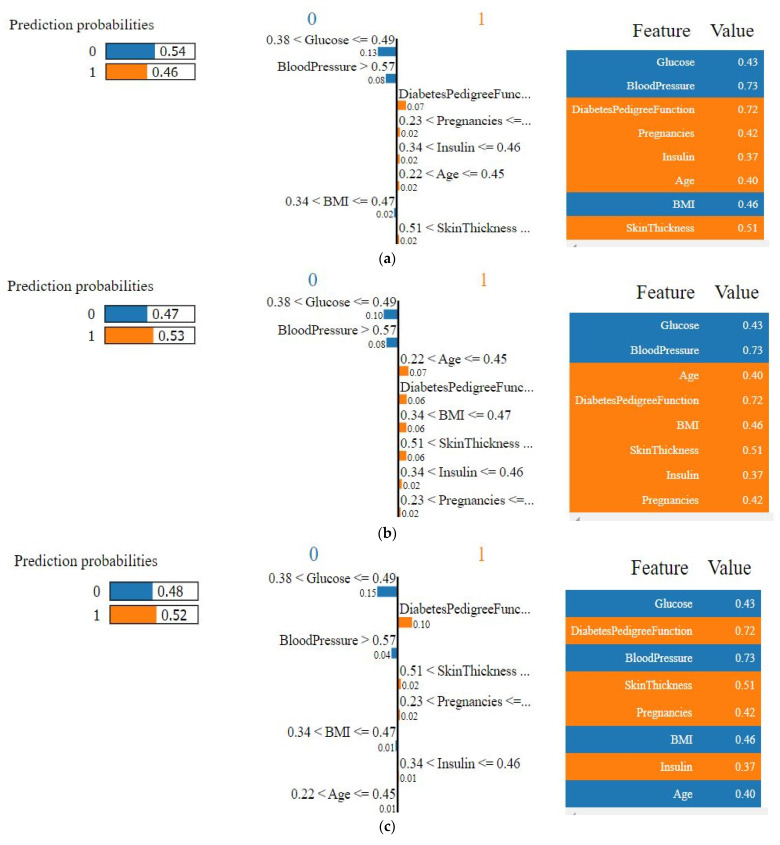
LIME tabular explainer of (**a**) ANN, (**b**) SVM, (**c**) LR, (**d**) RF, (**e**) ADA, (**f**) XGB, and (**g**) Voting classifier.

**Figure 7 sensors-22-07268-f007:**
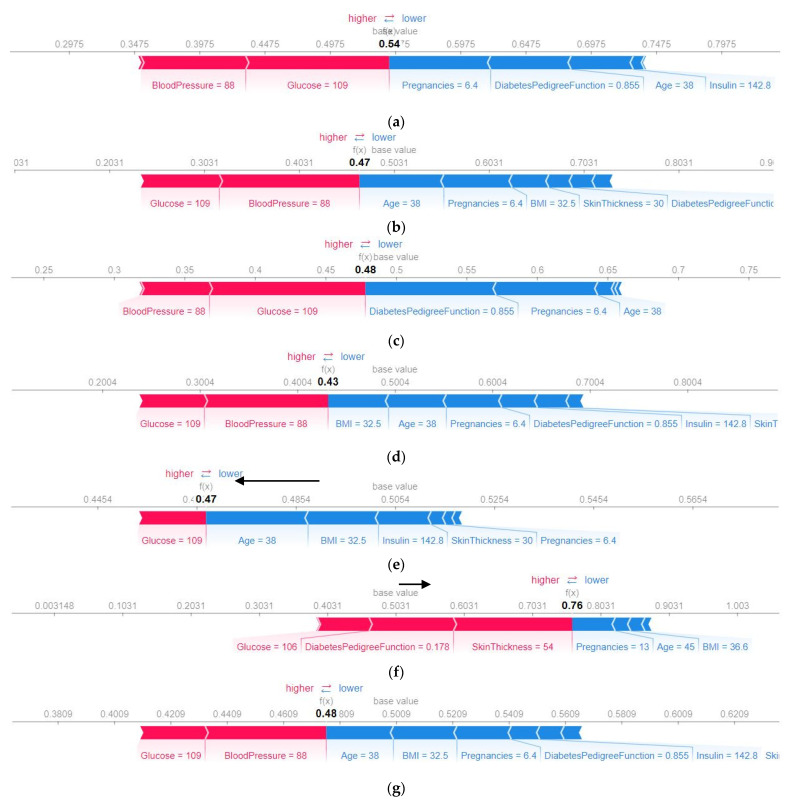
SHAP force plot of (**a**) ANN, (**b**) SVM, (**c**) LR, (**d**) RF, (**e**) ADA, (**f**) XGB, and (**g**) Voting classifier using the five-fold.

**Figure 8 sensors-22-07268-f008:**
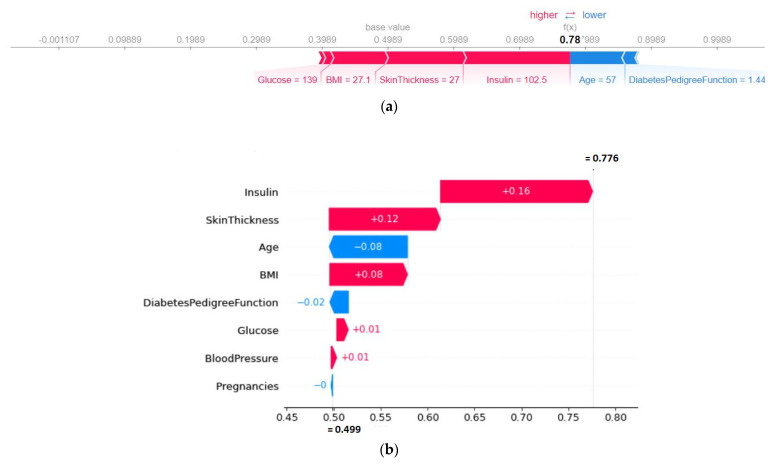
(**a**) SHAP force plot and (**b**) water plot for a test sample of the voting classifier using the five-fold.

**Figure 9 sensors-22-07268-f009:**
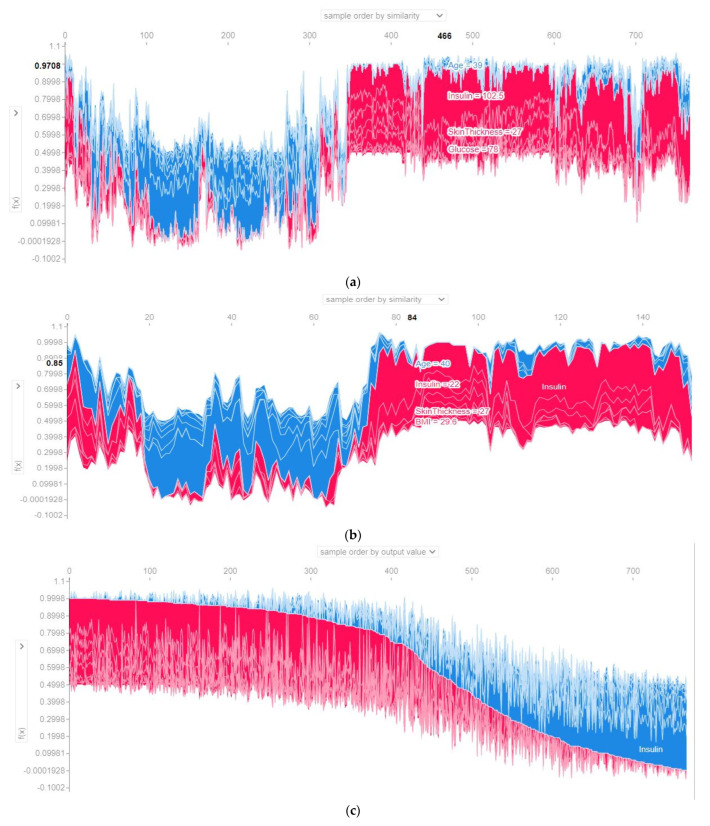
Output probability based on the sample order by the similarity of the voting classifier using (**a**) all folds (**b**) last fold and the sample order by the output value of the voting classifier using (**c**) all folds (**d**) last fold.

**Figure 10 sensors-22-07268-f010:**
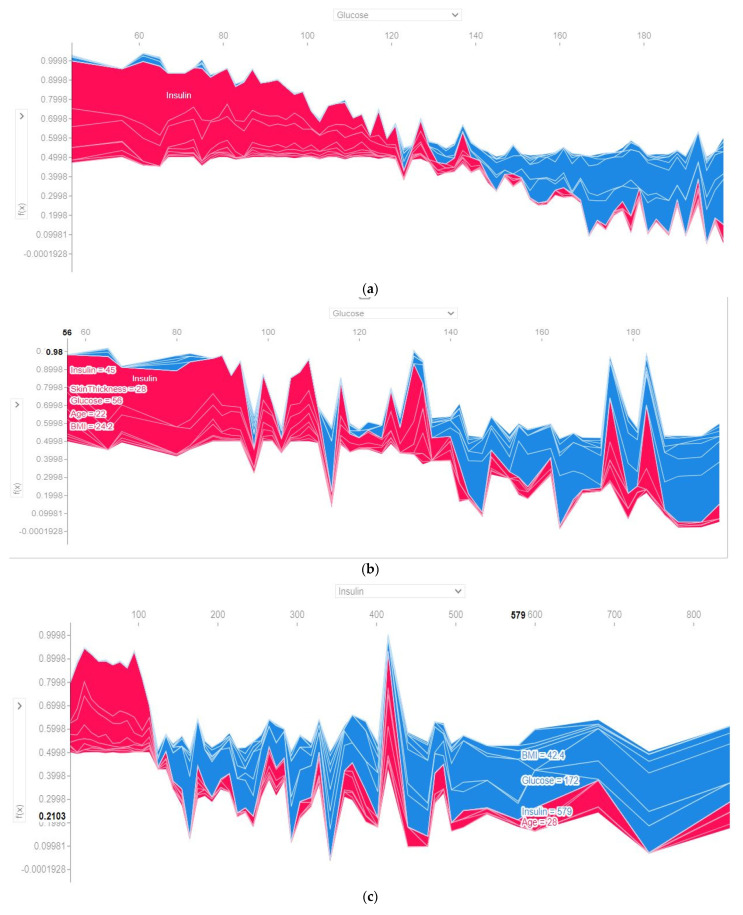
Output probability of glucose by the voting classifier using (**a**) all folds and (**b**) last fold and output probability of insulin by the voting classifier using (**c**) all folds (**d**) last fold.

**Figure 11 sensors-22-07268-f011:**
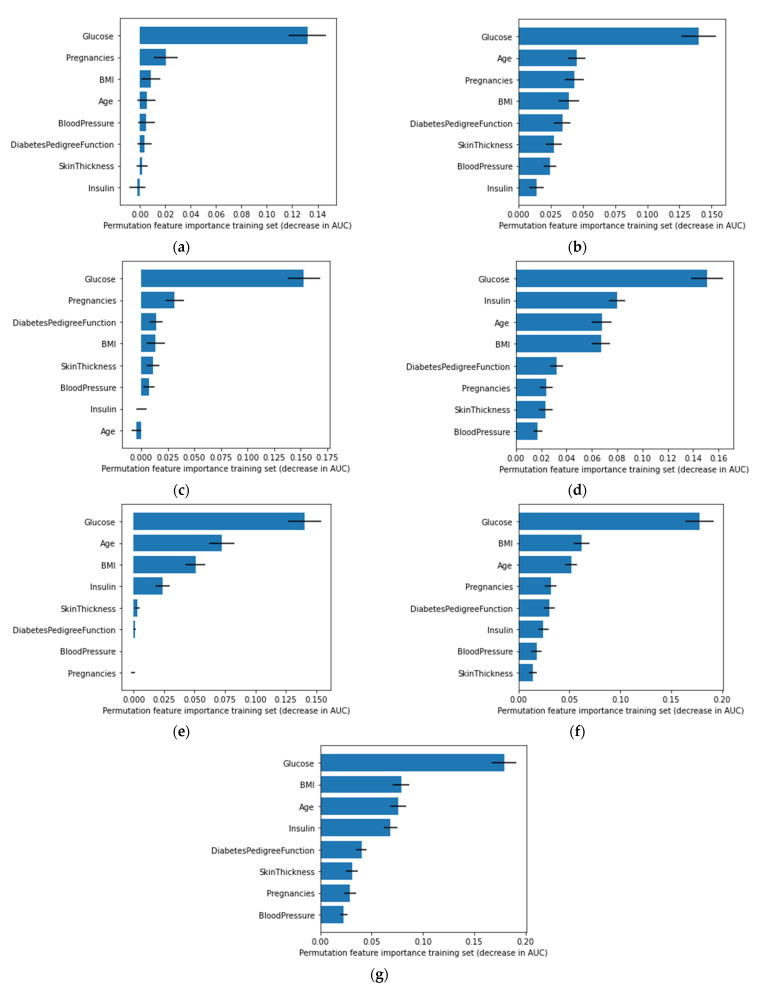
Permutation feature importance of (**a**) ANN, (**b**) SVM, (**c**) LR, (**d**) RF, (**e**) ADA, (**f**) XGB, and (**g**) Voting classifier.

**Figure 12 sensors-22-07268-f012:**
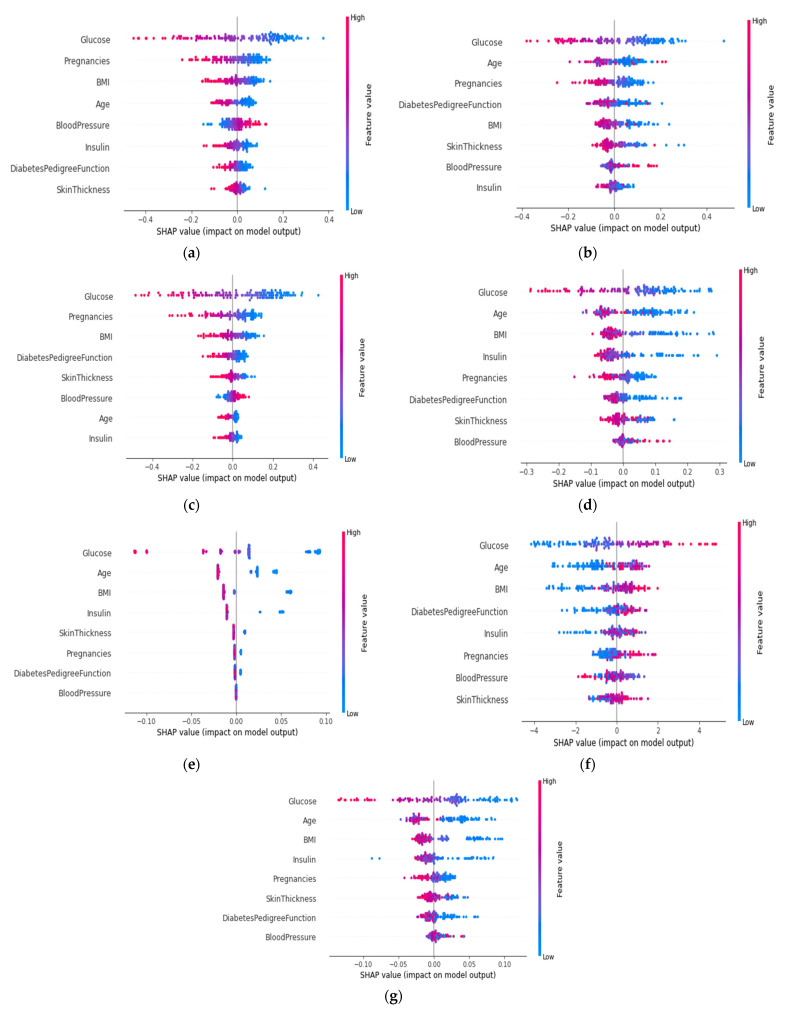
Summary plot of (**a**) ANN, (**b**) SVM, (**c**) LR, (**d**) RF, (**e**) ADA, (**f**) XGB, and (**g**) Voting classifier.

**Figure 13 sensors-22-07268-f013:**
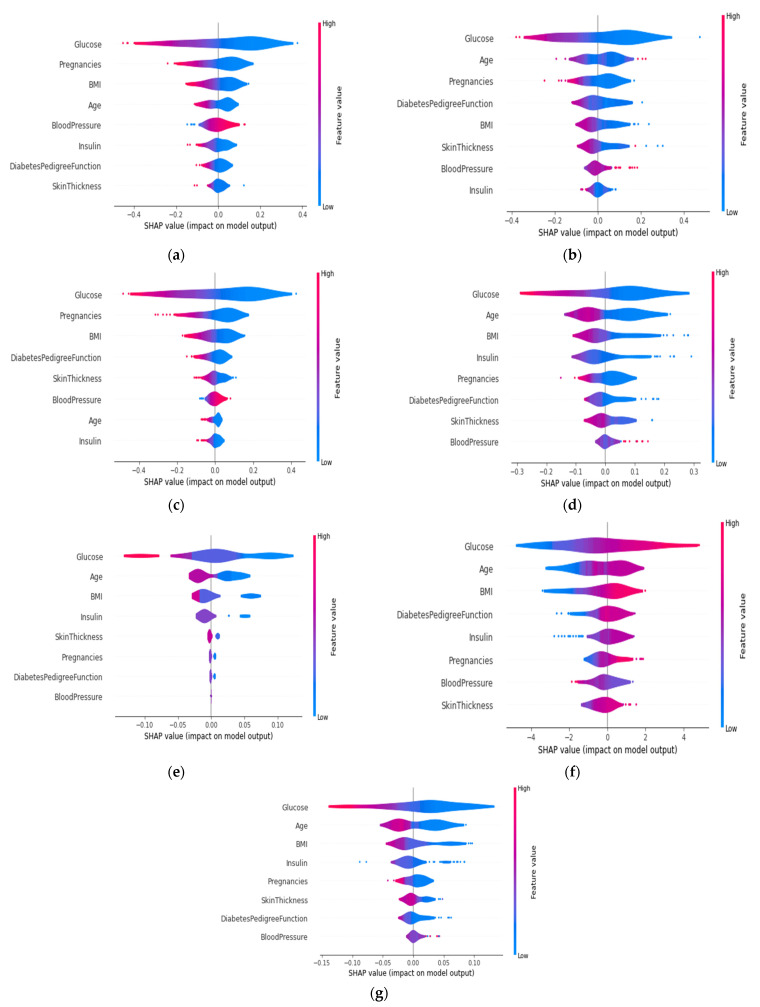
Violin distribution of (**a**) ANN, (**b**) SVM, (**c**) LR, (**d**) RF, (**e**) ADA, (**f**) XGB, and (**g**) Voting classifier.

**Figure 14 sensors-22-07268-f014:**
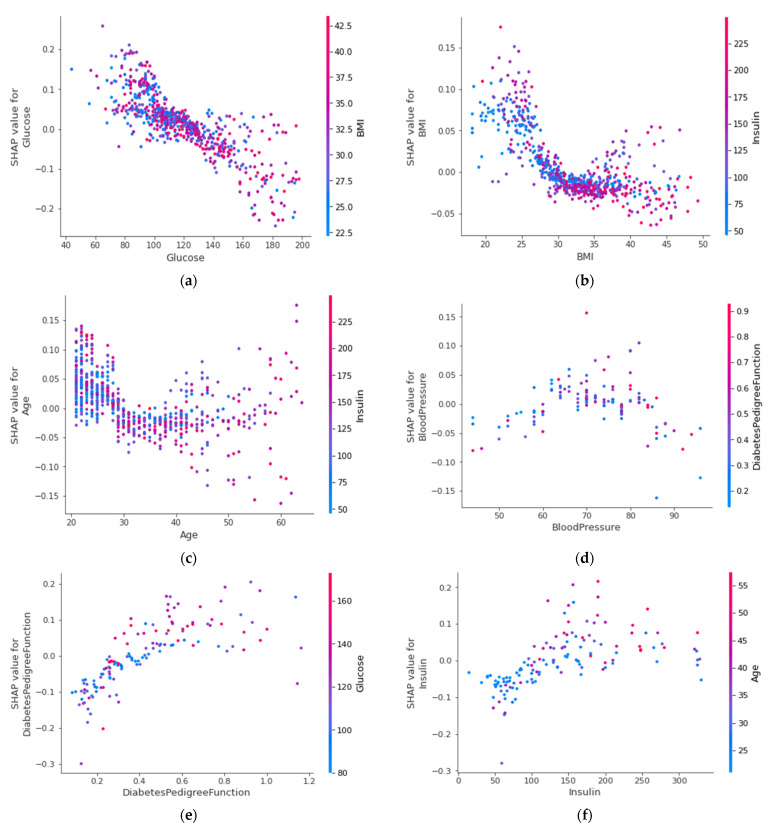
SHAP dependence plot by voting classifier for (**a**) glucose, (**b**) BMI, (**c**) age, (**d**) blood pressure, (**e**) diabetes degree function, (**f**) insulin, (**g**) pregnancies, and (**h**) skin thickness using all folds.

**Table 1 sensors-22-07268-t001:** Description of the attributes available in the Pima Indian Diabetes dataset.

Attribute	Attribute Type	Attribute Description
Pregnancies	Numeric	Number of times pregnant
Glucose	Numeric	Plasma glucose concentration (mmol/L) a 2 h in an oral glucose tolerance test
Blood Pressure	Numeric	Diastolic blood pressure (mm Hg)
Skin Thickness	Numeric	Triceps skin fold thickness (mm)
Insulin	Numeric	2 h serum insulin (mu U/mL): Insulin-resistant (IR) cells lead to prediabetes and type-2 diabetes. “2 h post glucose insulin level” is a cost-effective, convenient, and efficient indicator to diagnose IR [29,30]
BMI	Numeric	Body mass index weight in kg/(height in m)
Diabetes PF	Numeric	Diabetes pedigree function: indicates the function which measures the chance of diabetes based on family history.
Age	Numeric	Age (years)

**Table 2 sensors-22-07268-t002:** Statistical description of the Pima Indian Diabetes dataset.

	Pregnancies	Glucose	Blood Pressure	Skin Thickness	Insulin	BMI	Diabetes- Pedigree Function	Age	Outcome
count	768	768	768	768	768	768	768	768	768
mean	3.84	121.59	72.37	29.11	153.18	32.42	0.47	33.24	0.34
std	3.36	30.49	12.2	9.42	98.38	6.88	0.33	11.76	0.47
min	0	44	24	7	14	18.2	0.07	21	0
25%	1	99	64	23	87.9	27.5	0.24	24	0
50%	3	117	72	29	133.7	32.09	0.37	29	0
75%	6	140.25	80	35	190.15	36.6	0.62	41	1
max	17	199	122	99	846	67.1	2.42	81	1

**Table 3 sensors-22-07268-t003:** Number of classes before and after using the SMOTETomek on the training dataset.

	Before the SMOTETomek	After the SMOTETomek
Numbers in class 0 (non-diabetic)	400	393
Numbers in class 1 (diabetic)	214	393

**Table 4 sensors-22-07268-t004:** Optimal hyperparameters used in the algorithms.

Algorithms	Optimal Parameters
Artificial neural network	Batch size = 5, epochs = 20
Support vector machine	default
Logistic regression	C = 10
Random forest	default
XGBoost	number of estimators = 20
AdaBoost	number of estimators = 300, learning rate = 0.01

**Table 5 sensors-22-07268-t005:** All models’ performance for the five-fold cross-validation of the dataset with the balanced classes.

Algorithms	Precision	Recall	F1-Score	AUC Score	Accuracy	
					Train	Test
ANN	0.77	0.78	0.78	0.87	0.81	0.79
SVM	0.79	0.81	0.80	0.88	0.87	0.79
LR	0.78	0.79	0.78	0.86	0.80	0.78
RF	0.87	0.88	0.87	0.94	1.00	0.88
XGB	0.88	0.89	0.88	0.92	0.99	0.88
Ada	0.82	0.85	0.83	0.95	0.86	0.83
Voting Classifier (XGB + RF)	**0.88**	**0.89**	**0.89**	**0.95**	**0.99**	**0.90**

**Table 6 sensors-22-07268-t006:** Comparison of the diabetes detection model outcomes with the previous studies.

Approach	Train Test Split	Result (%)				Ref.
Decision tree Random forest Naive Bayes	70:30 train test ratio		DT	RF	NB	[13]
		Accuracy Precision Sensitivity Specificity F1 score AUC	74.78 70.86 **88.43** 59.63 78.68 78.55	79.57 **89.40** 81.33 75.00 85.17 86.24	78.67 81.88 86.75 63.29 84.24 84.63	
RF AdaBoost Soft voting classifier	70:30 train test ratio		RF	Ada	Voting classifier	[10]
		Accuracy Precision F1 score Recall AUC	77.48 71.21 64.38 58.75 78.10	75.32 68.25 60.13 53.75 74.98	79.08 73.13 71.56 70.00 80.98	
RF	Not mentioned		RF	ANN	K mean clustering	[2]
		Accuracy AUC	74.70 80.60	75.70 81.60	73.60 -	
ANN XGB	Not mentioned			ANN	XGB	[12]
		Accuracy Sensitivity Specificity AUC		71.35 45.22 85.20 65.00	78.91 59.33 89.40 88.00	
Naive Bayes SVM DT	10-fold Cross-validation		NB	SVM	DT	[11]
		Precision Recall F1 score Accuracy	75.9 76.3 76 76.3	42.4 65.1 51.3 65.1	73.50 73.80 73.60 73.82	
Proposed soft voting classifier (XgBoost + RF)	5 fold Cross-validation	Accuracy Precision Recall F1 score AUC			**90** 88 **89** **95** **95**	-

**Table 7 sensors-22-07268-t007:** Values (Natural unit) of every feature of a selected test sample corresponding to Figure 6.

Features	Values (Natural Unit)
Glucose	109.00
Blood pressure	88.00
Insulin	142.80
Skin thickness	30.00
Pregnancies	6.40
BMI	32.50
Diabetes pedigree function	0.85
Age	38.00

## Data Availability

The data presented in this study are available in the article.

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
