# Peer review of "An Ensemble Approach for the Prediction of Diabetes Mellitus Using a Soft Voting Classifier with an Explainable AI"

_sensors, 2022, doi:10.3390/s22197268_

Round 1

Reviewer 1 Report

Authors need to add the subsections in the introduction section: “Research Challenges”, “research motivation”, “Problem Statement”.

How the proposed research filled the gap? Need a brief summary at the end of literature section.

The proposed work should be compared with recent published works using the same dataset. Need to add figure/table for this comparison.

The explainable AI part seems interesting. However, the figures are low quality/not visible. i.e., Figures 9, 10, 11, 12. Author needs to redraw the clear figures with high quality.

Figures 14 and 16 show a chunk of features from the dataset to show the impact. Why and how these specific features are selected to show the exploitability? Is it difficult to show all features in this analysis? Need strong reasons.

Reviewer 2 Report

Authors considered very important problem of diabetes diagnostics. It was stated that the main difference is explainability of proposed algorithm. This assumes that clinician can understand why model diagnose/not diagnose diabetes. Unfortunately authors forgot to explain most of their decisions in presented paper.

Abstract. It is interesting but abstract does not contain AI at all.

1.     Line 22. “Here, eight machine learning algorithms have been developed”. It looks like inaccurate. I’ll read further but I think that 8 algorithms were used but not developed. I am correct – see line 75.

2.     Line 23. “explanations have been explained” seems far from good practice. Maybe “explanations have been” “produced” or “proposed” or “presented” something like this.

3.     Line 24. “Shapley Additive explanations” is very expansive (in accurate implementation”) and frequently has very far relations with real relations between inputs and outputs. Review and some examples of issues related to Shapley values can be found in “Fryer, D., Strümke, I. and Nguyen, H., 2021. Shapley values for feature selection: the good, the bad, and the axioms. IEEE Access, 9, pp.144352-144360.”

4.     Line 25. “The accuracy of the developed weighted ensemble model was 80% using a five-fold cross-validation.” This statement cause three questions.

4.1.  The first question is what do you call accuracy? If you use 1 – misclassification rate for so unbalanced classes then it is not reasonable to continue reading. If you used something else (balanced accuracy, F1 score, etc.) it should be implicitly stated.

4.2.  The second question is about accuracy of previously known models: do all previously developed models have worse accuracy? May be some of such problems have accuracy about 90%? In this case your model will be incomparable worse. Please present accuracy of previously developed models.

4.3.  The third problem is related to reproducibility of your model. It is well known, that cross validation approaches are reasonable for reproducible models only (see, for example, Hastie, T., Tibshirani, R., Friedman, J.H. and Friedman, J.H., 2009. The elements of statistical learning: data mining, inference, and prediction (Vol. 2, pp. 1-758). New York: springer.

5.     Line 26. “K-nearest neighbors (KNN) was used for the imputation”. In this form it looks like unclear. The best way is to add something like “with k =3” .

6.     Line 28. Statement “to balance the dataset” is inaccurate. Do you mean “to balance classes in the new dataset”? There are two important moment: first of all you balance classes but not dataset and the second after balance of classes you have another dataset. This new dataset is not “Pima Indian diabetes dataset” and all comparison and estimations must be carefully corrected.

7.     Line 31. Statement “at the very early stage of diagnosis.” Is very strange or inaccurate. Do you mean “at the very early stage of disease”?

Introduction

8.     Line 36. Statement “Diabetes has recently become one of the top causes of death in developing countries” is inaccurate. Diabetes is almost never is cause of death. Usually there are other causes, like stroke, heart failure, etc. Please be careful in statements.

9.     Line 37. Statement “to find a solution for this critical disease” is at least strange. Do you think that euthanasia is solution? Do you mean treatment? Be careful in word selection.

10.  Line 38-39. Your description is correct for type 1 diabetes. Type 2 diabetes (more than 80% of all diagnosed diabetes) can be insulin independent. Please be more accurate.

11.  Fragment in lines 36-45 must be rewritten in more accurate and exact statements.

12.  Line 50. Statement “Diabetes is caused by the change in glucose levels in the body.” Is wrong. Diabetes can cause glucose level in body. Changes of glucose levels in the body cannot cause diabetes.

13.  Line 53. “Other laboratory tests…” Usage of word “other” assumed that something was presented before. For example, I like apples but do not like other fruits. In this case I have opposition of apple and other fruits. Since there were no described tests before it is not clear what authors mean.

14.  Statement “Patients with type 2 diabetes require life-saving insulin” is exactly wrong. There are (according to some sources it is more than 50% of all type 2 diabetes cases) insulin independent type 2 diabetes cases.

15.  The second paragraph (exclude the last three phrases) must be rewritten in more accurate style.

16.  Line 60. Fragment “for disease prediction” seems inaccurate. Maybe something like “for disease diagnostics”

17.  Line 61. “different types of disease” must be “different types of diseases”.

18.  Line 63. “method can accurately detect diabetes at an early stage.” What does it mean “accurately”? What are sensitivity and specificity?

19.  Line 63. “For diabetes prediction” is inaccurate. Word “prediction” usually means that we try to define what will be in future. For example, “patient has 70% chances to have diabetes in half of year”. If we define (identify, etc.) has patient diabetes now or not it is problem of diagnostics but not prediction. It is very important to use correct terms.

20.  Line 66. “Some got poor accuracy”. Please cite. Please present this poor accuracy. For differential diagnostics accuracy 80% frequently considered as not very good.

21.  Line 68. “explanations behind the decision were not described adequately”. Please cite.

22.  Line 68. “Due to the poor accuracy and explainability”. Please cite. Now I should state that background is not described at all. We know nothing about diabetes diagnosis before this paper.

23.  Line 80 “for the doctors to understand and implement the model.” Be accurate. Doctors or any other clinician can use models but will not implement them.

Literature review

24.  Literature review written in informal and inaccurate style. Authors forgot to define accuracy but widely used this term. As a result many statement are about nothing.

25.  Line 105. Citation should be at the beginning of fragment, which refer to this source. I think that it should be after “Other researcher” in line 104. Author stated that 78.2% is poor accuracy but their (authors) 80% accuracy is good. It is unclear, why 1.8% considered as vary big changes of accuracy. Since [11] used SVM, this model allow us directly estimate influence of each input feature and explain decision. This was omitted.

26.  Line 108. “Outliers were removed”. It is not clear, why outliers should be removed. In many cases removing of outliers is not appropriate procedure.

27.  Line 109. Statement “data distribution was balanced” is inacceptable. There is no definition of balanced or imbalanced distribution. Distribution it is not dataset. What do you think, are normal or Bernoulli distributions balanced?

28.  Line 110. Authors forgot to say which of these four databases was Pima 104 Indian dataset. I think that it was the first one and proposed in [12] solution has accuracy 96%. This means that model with accuracy 80% should be excluded from consideration.

29.  Line 114. “Ensemble technique was also introduced by Kumari et al. [13],”. Paper [13] is issued in 2021 but ensemble technique was invented at least 50 years before. Be accurate.

30.  Line 115. Statement “for classifying diabetes mellitus” is inaccurate. There are at least three types of diabetes. Do you mean recognition type of diabetes? Do you mean diagnosis of diabetes? These two problems are absolutely different.

31.  Line 116. “using a breast cancer dataset, which provided an accuracy of 97%.” This statement is something very strange, because of dataset cannot provide any accuracy. Please be more accurate.

32.  Lines 120-124. “Kibria et al. [5] found an accuracy of 83% in diabetes detection using logistic regression (LR) and KNN algorithm was employed for imputation of missing values. By removing the outliers and balancing the data distribution, opportunities can be created to increase prediction performance.” I completely confused: [5] provided 83% of accuracy but they do not removed outliers. In current paper outliers were removed and accuracy increased up to 80%. What does it mean? It should be clearly described in text. I should say, that logistic regression has implicit mechanism of global and local estimation of feature informativity (influence, etc.).

33.  Line 125. “Three machine learning algorithms were built using the Pima Indian dataset”. It is completely wrong statement: algorithm cannot be built by using any dataset. Model can be built by ML algorithm for some dataset. Be accurate.

34.  Line 127. “data was imbalanced” is widely used but is very inaccurate. Classes can be imbalanced but not data.

35.  Line 130. “Balancing the dataset and removing the outlier will help the model to predict more accurately.” To remove outliers it is necessary to prove that these outliers are caused by errors or mistake. Removing outliers because they are cause error for your model is simply falsification of results. Let us call outlier all cases where our model produces wrong recognition. It is very important that outlier can be defined only with respect to some model. Why to not use our model instead of some “isolation forest” which assumed special type of data distribution and we do not know satisfied our data these assumptions or not. Then we can remove them (all cases with wrong recognition) and we will have model with 100% accuracy. Please be more reasonable. Please check, can be proposed outlier detection technique used to your dataset? Have we any reason to remove outliers? I should stress, that it should be reasons instead of wanting.

36.  Line 137. What do you mean “fused”? Do you mean “used”?

37.  Line 137. What do you mean “The result of these models became the fuzzy model's input membership function”? Do you mean “output membership function”?

38.  Line 140. What do you mean “However, the ML model was not interpretable”? Linear SVM is clearly interpretable. ANN can be interpretable (see, for example, https://arxiv.org/abs/2005.06284, Evgeny M Mirkes Artificial Neural Network Pruning to Extract Knowledge). Please be more accurate in your statements.

39.  Paper [18] demonstrates accuracy 98% and it is worse than 80%. It is necessary to explain why do you prefer 80% accuracy method if you have 98% method?

40.  Line 172. “Figure 1Figure 1.”

41.  According to Figure 1, changing of dataset with class balancing was produced for each training set separately. As a result, it is not 5 fold CV method.

42.  According to chapter 3 and figure 1 following ML methods was used Random forest, AdaBoost, ANN, DT, XGBoost, SVM, LR. Part of these methods are reproducible: AdaBoost, and XGBoost; part are irreproducible: Random forest and ANN; and DT is special case. Usage of cross validation for irreproducible methods is absolutely unreasonable (see, for example, Elements of statistical learning). As I wrote above, DT is special case because the small change of training set can completely change topology of tree. Meaning of CV accuracy for DT is meaningless.

43.  The finally selected set of algorithms include two irreproducible ones: Random forest and ANN. For these algorithms usage of cross validation is meaningless and it is absolutely unclear what authors stated as 80% accuracy.

44.  There is no reference for dataset source. It is absolutely inacceptable.

45.  According to Table 2 there is no missing values in dataset. It is absolutely trash and table can be removed. It is absolutely clear that features “Glucose”, “Blood Pressure”, “Skin Thickness”, “BMI” cannot have value 0. I am not sure in feature Insulin, because I do not know meaning of this field: amount of insulin in blood or amount of injected insulin. In the first case value zero is inacceptable but in the second it can be correct. What is meaning of zero value in feature “number of pregnancies”? Do we have only female in this database? What is way to encode the number of pregnancies for male?

46.  Figure 3. Absolutely wrong caption. It should be “Heatmap of Pearson’s correlation coefficients for all diabetes features”. It is unclear how PCC were calculated: for complete records, for all complete pairs of two attributes, in any other way?

47.  Line 214. “target variable and independent features were correlated” is absolutely wrong statement. I think that authors mean “the correlation between target variable and independent features were measured by…”.

48.  Meaning of features is not described. For most of them, meaning is more or less clear but for value “insulin” it is not clear.

49.  Figure 4. The bottom line is inacceptable because of left and right graphs have different scale in y-axis. Graphs produces wrong impression.

50.  It is unclear how were created graphs in figure 4: for known values, or for all records include zeros.

51.  Line 238. Statement “of any of the hormones” is something strange because of only one hormone is presented in this database – insulin.

52.  Lines 237-242. Method of distance calculation is not presented. There are two different approaches – search neighbours in set of complete records only or use special way to calculate distance between incomplete records. Outcome of these method is slightly different.

53.  Figure 5 present something strange. It is exactly not a distribution. If I correctly guess than we have 673 records after removing outliers. This means that 96 records (more than 10%) were removed. If I correctly guess that left graph present number of known values before imputation and the right graph contains number of known values after imputation. The right graph can be removed because of it is constant. According to graph authors considered 0 values of pregnancies as missing values and impute it. Since we have ages from 21 it is very strange, because of most of 21 women never were pregnant.

54.  Data MUST be CAREFULLY described. This description must contain description gender of patients, meaning of each feature and so on.

Reviewer 3 Report

This publication has a certain scientific and practical significance and contributes to the development of deep learning methods. I think it will be useful for practical application in detecting diabetes in complicated cases.

Round 2

Reviewer 1 Report

My comments are addressed. The revised version seems technical and easy to understand. 

A few research limitations should be added in the conclusion section. 

Reviewer 2 Report

New version of text is essentially better than the original one.

Point 4. Authors discussed slightly SHAP in 3.4.10 but they completely omitted very important questions for this methodology: Does result of SHAP have any relation to real regularity? This question partially is discussed in [42]. The second reason to not use SHAP is huge time consumption. And the third and most important reason is existence of explicit methods of calculation of influence of each input feature to result (the same as explainability) for ensemble learning, ADABoost, Decision tree, XGBoost, Logistic regression, SVM and ANN. It is absolutely unclear why usage of some artificial tool (SHAP) can produce more “realistic” explanation then native methods? This problem is not discussed.

In the correction of point 7 there is inaccuracy. Statement “to balance the target class of the dataset” is incorrect. You can balance “classes in dataset” but not “target class”. You also forgot to add to text that test set was not balanced.

Point 25. In response authors wrote “Accuracy is a very common term in data science”. It is true but also true the following statement: there are many quality measures which commonly called accuracy. As a result it is absolutely unclear what is accuracy. Is it accuracy as 1 - misclassification rate? Is it balanced accuracy? Is it sensitivity? Wikipedia presents (https://en.wikipedia.org/wiki/Sensitivity_and_specificity) list of 21 “accuracies”. This means that we should each time exactly specify used accuracy. Otherwise, presented numbers become meaningless.

Line 219. Statement “Due to the poor accuracy” is strange because accuracy 98% is greater than authors’ 80%. In [16] SVM provide 80% accuracy. SVM can be easily interpreted. Do authors mean that they select models by three criteria: good performance, high explainability and usage of cross validation for performance estimation? I think it will be very good to state it clearly instead of “Due to the poor accuracy”. I completely agree with authors that hold out performance estimation can be very different for the same dataset and the same fraction of training set. You can find discussion of this problem in Hastie, T., Tibshirani, R., Friedman, J.H. and Friedman, J.H., 2009. The elements of statistical learning: data mining, inference, and prediction (Vol. 2, pp. 1-758). New York: springer.

Line 312. “In KNN, each sample's missing values are imputed using the mean value from the dataset's n neighbors nearest”. If you use n nearest neighbours then you should refer to nNN. Please be more accurate.

Point 27 Line 108 and point 36 line 130. “Outliers were removed”. It is not clear, why outliers should be removed. In many cases removing of outliers is not appropriate procedure. Response 27: The reason has been explained in section 3.3.1 and Section 4.2 (near Figure 8). Section 3.3.1 contains description of outlier identification but not reasons to remove them. Section 4.2 contains explanation why authors removed outliers: because accuracy for dataset without outliers is greater. In this case we can remove all wrongly recognised records and accuracy become essentially better. As I wrote before there are at least two types of outliers: errors and atypical. Errors can be removed but atypical cases must be used or you should suggest method of this atypical cases recognition and create another model for these atypical records. Current approach seems is not honest: this record is difficult and I will not process it.

Point 42. Your answer is not answer at all. If you apply SMOTETomek for each of 5 training sets separately, then it is not cross validation. If you isolate all five folds (let us call them F1,…,F5), then applied SMOTETomek and produce other (balanced) folds f1,…,f5. If you will use for fold k training set as union of fi (i is not k) and test set Fk then it will be cross validation. Please describe your testing protocol in details, accurately and using exact terms.

Points 43 and 44. Unfortunately, authors does not completely understand this point. Problem with irreproducible (non-robusr, non stable, etc.) models only partially related to stochasticity of model fitting process. Result of irreproducible methods can drastically depends on small changes of dataset or even on order of data records in fitting process. For example, DT has no any stochastic procedure in fitting but removing of one record can completely change topology of tree. This is reason to not use cross validation for DT.

Point 45. Proposed link to data is not worked. It will be very useful if you can provide other link if it is possible.

Point 47. Your answer is incomplete. Did you calculate PCC after data imputation? If yes, please state it. If not then you should describe how you select subsample to calculate PCC.

Figure 4. It is very interesting, but glucose does not have any outliers before outlier detection and have several after removing. BMI also have outliers after outlier removing. This effect was not discussed. Does it mean that we must use outlier removing recursively?

Point 53. Euclidean distance can be used to calculate distances between COMPLETE points (all coordinates are known) and for points with PARTIALLY UNKNOWN coordinates. What approach did you use?

Figure 5. The right figure is absolutely meaningless because of constant. Pleas remove right figure.
Tables 5 – 12 can be moved to appendix. I think that in text will be more useful table which contains the last line of tables 5-12 but for all methods. The same for tables 13 and 14.

Figure 6. Graph for ROC curve MUST be square. It is standard requirement. You also can remove title “Receiver operating characteristic example”. At least word example MUST be removed. The same figure 7.

Section 4.4. In lime you use some intervals of values for each attribute. These intervals are not described. You should explain selection of intervals. I am not sure that there are common intervals for all world but I know that in UK there are following intervals for BMI: BMI<18.5 is Underweight, from 18.5 to 25 is normal, from 25 to 30 is overweight, from 30 to 35 is obesity class 1, from 35 to 40 is obesity class 2, BMI>40 is obesity class 3. Used in paper BMI from 0.34 to 0.47 is absolutely meaningless for clinicians. If you want to present data to clinicians, then all values MUST be in natural units and intervals should be taken from medical standards. For your paper, I think you should use Indian standards. Figure 9 must be completely redrawn with usage of natural units: years for age, and so on. You can apply any normalisation to fil model but you MUST convert data back to natural units for clinicians. I found that for figures d, e, g, and h you presented natural units. It should be done for all. What is reason to use interval from 28.7 to 32.8 for BMI?

Figure 9. I think it is reasonable to split this figure to several figure. I can say that “Prediction probabilities” are partially correct for logistic regression only. For ANN, SVM, ADABoost, and XGBoost it is completely wrong. For RF it can be correct but I do not know method of these values calculation in Lime.

Line 365 “is a higher possibility that”. Possibilities can be wider but I think that “probabilities” should be used here.

Figure 10. What is meaning of horizontal axes? Is it probability of diabetes?

Figure 10. How I can find numerical value of influence for each attribute?

Line 676. What does it mean “The base value is the average model output for all test data if any feature is not known for the current prediction” for database after imputation?

Line 695. “Figure 11 illustrates the supervised clustering of all cases according to their similarities, output values, and features.” What type of similarity is used in this context. There are huge variety of similarities and which similarity you used in this case must be declared explicitly.

Line 698. According to figure 5 there are 673 records. Why you tested 650 records only?

Conclusion. I am not sure that most of clinicians can understand SHAP explanation (after figure 10) because of reading of these graphs is difficult even for computer science people but you can try to explain them to clinicians.
